# Relationship between Cardiac Acoustic Biomarkers and Pulmonary Artery Pressure in Patients with Heart Failure

**DOI:** 10.3390/jcm11216373

**Published:** 2022-10-28

**Authors:** Tetsuya Kaneko, Atsushi Tanaka, Kota Jojima, Hisako Yoshida, Ayumu Yajima, Machiko Asaka, Nobuhide Yamakawa, Tomoyuki Kato, Norihiko Kotooka, Koichi Node

**Affiliations:** 1Department of Cardiovascular Medicine, Saga University, 5-1-1 Nabeshima, Saga 849-8501, Japan; 2Department of Medical Statistics, Osaka Metropolitan University Graduate School of Medicine, 1-4-3 Asahi-machi, Abeno, Osaka 545-8585, Japan; 3Corporate R&D, Asahi Kasei Corporation, 1-1-2 Yurakucho, Chiyoda-ku, Tokyo 100-0006, Japan; 4Budounoki Medical Clinic, 2-21 Mizugae, Saga 840-0054, Japan

**Keywords:** acoustic cardiography, cardiac acoustic biomarkers, heart failure, heart sound

## Abstract

Since an elevation of pulmonary artery pressure (PAP) often precedes clinical worsening of heart failure (HF), early and non-invasive detection of this sign is useful in HF care. This study aimed to assess whether cardiac acoustic biomarkers (CABs) are associated with the elevation of PAP in patients with HF. Patients with HF scheduled to undergo right heart catheterization were prospectively enrolled. CABs were concurrently recorded during catheterization at rest (baseline) and while applying a handgrip (exercise). Forty-nine patients were included in the analysis, and their mean PAP significantly increased after exercise compared to baseline. Several CABs correlated significantly with mean PAP by absolute values, among which S2 Width (*r* = 0.354; *p* = 0.014 and *r* = 0.363; *p* = 0.010) and S3 Strength (*r* = 0.375; *p* = 0.009 and *r* = 0.386; *p* = 0.007) were consistent throughout baseline and exercise. The response of CABs to exercise-induced PAP elevation was divided into two patterns: increasing and decreasing. The frequency of cardiac index below 2.2 mL/m^2^ was significantly higher in the decreasing pattern. CABs related to S2 and S3 showed significant correlations with absolute PAP values both at baseline and after exercise in patients with HF, but no significant correlations between their changes from baseline to post-exercise were observed in this study population. Further research is therefore needed to assess whether CABs can sensitively reflect changes in PAP according to HF status and underlying phenotypes.

## 1. Introduction

Heart failure (HF) is a chronic and progressive condition with a poor long-term prognosis because of recurrent acute exacerbations that result in rehospitalizations [1]. Patients with HF who have hemodynamic deterioration due to left ventricular diastolic dysfunction tend to develop elevated pulmonary artery pressure (PAP) several weeks before the onset of clinical symptoms [2]. Interventions based on long-term monitoring of PAP obtained from implantable devices have shown efficacy in preventing events (such as hospitalizations) that worsen HF and are expected to deliver a new strategy in the management of patients with HF [3].

For a long time, physicians have examined the hemodynamic condition of patients with cardiovascular diseases with the aid of auscultation of abnormal heart sounds from the body surface. Currently, the abnormality of heart sounds can be parameterized by computerized algorithms, which are increasingly being used to monitor the condition of patients outside hospitals and to predict deterioration [4,5]. Such parameters are described as cardiac acoustic biomarkers (CABs) [6] derived from noninvasively measured heart sounds and electrocardiograms obtained from acoustic cardiography [7]. Several HF studies have previously shown that CABs were associated with an increased left ventricular filling pressure and left ventricular dysfunction [8,9,10]. In addition, CABs reportedly have possible application in the diagnosis of pulmonary artery hypertension since they showed an association with mean PAP and pulmonary vascular resistance (PVR) measured at rest during right heart catheterization in patients with pulmonary artery hypertension [11]. The authors previously applied the diagnostic technology to the general population of patients with pulmonary hypertension, differentiating its phenotypes by referring to parameters of right heart catheterization at rest [12]. However, the relationship between CABs and a dynamic change in PAP has not been evaluated in patients with HF.

The present study aimed to evaluate the feasibility of the noninvasive technology in monitoring the elevation of PAP by analyzing the relationship between CABs and concurrently recorded PAP during right heart catheterization at rest and after exercise in patients with HF.

## 2. Materials and Methods

### 2.1. Subjects

This was a single-center, prospective observational study of the relationship between PAP and CABs, conducted at the Saga University hospital from February 2020 to September 2021. CABs were recorded during right heart catheterization in 50 consecutive patients with HF who were diagnosed according to the Framingham criteria for congestive HF and had stable hemodynamic conditions. The inclusion criteria were as follows: (1) men and women aged ≥ 20 years, and (2) patients with chronic HF admitted for right heart catheterization or patients with acute HF who had improved pulmonary congestion and were declared stable by a physician after treatment. The exclusion criteria were as follows: (1) presence of severe aortic stenosis; (2) presence of degenerative mitral regurgitation; (3) post-mechanical valve replacement; (4) use of assistive circulatory device; (5) concurrent malignancy; and (6) pregnancy, possible pregnancy, or lactation. In addition to basic information such as age and sex, examination results of blood tests, echocardiography, 12-Lead electrocardiogram (ECG), and chest X-ray performed from a period close to the dates of right heart catheterization were collected and used to describe the baseline characteristics of patients. The study protocol was reviewed and approved by the ethics committee at Saga University, Japan. The study was conducted in accordance with the Declaration of Helsinki and all applicable laws and guidelines in Japan. Patients provided written informed consent before their enrollment in the study. This study was registered with the UMIN Clinical Trials Registry (ID: UMIN000039787).

### 2.2. Data Collection

#### 2.2.1. Right Heart Catheterization

Right heart catheterization was performed using a 6-French triple lumen Swan-Ganz catheter through the internal jugular or femoral vein under ultrasound guidance. The workflow of catheterization with simultaneous ECG and phonocardiogram (PCG) recording is shown in Figure 1. Baseline measurements included right arterial pressure, right ventricular pressure, PAP, pulmonary capillary wedge pressure (PCWP), and cardiac output at rest. PCWP was recorded with a Swan-Ganz catheter brought into the inferior lobar branch of the right pulmonary artery and wedged with dilated balloon. Handgrip was performed after baseline measurement. Patients were asked to squeeze a cylindrical grip at about 75% maximal voluntary handgrip contraction until exhaustion or the occurrence of some symptoms. They were further instructed to keep breathing to avoid Valsalva load. PAP and PCWP were measured near the end of the handgrip which may have been maximal load. After the exercise, post-exercise (cool-down) PAP was recorded. Cardiac output was calculated using the Fick method with the baseline mixed venous and artery oxygen saturation, and the cardiac index was calculated by standardizing cardiac output with the body surface area. Only cardiac index at baseline and PAP and PCWP at baseline and after exercise were used in the analysis in this report, and they are indicated as filled circles in Figure 1.

#### 2.2.2. CABs

The waveforms of PCG and ECG were synchronously recorded using a Holter ECG recorder device (AUDICOR AM-RT, Inovise Medical Inc., Portland, OR, USA). The device was attached to the patient’s chest before right heart catheterization, and the recordings were taken continuously during the examination. The electrodes were placed at the right and left upper and left lower chest areas, the third left intercostal space (3 L), and apex (V4) for the ECGs, whereas those for PCGs were obtained from accelerometers mounted on 3 L and apex electrodes.

Signals of PCG recorded at the apex and those for ECG were divided into 10-second sections. In each section, the heart sound categories, i.e., first, second, third, and fourth (S1, S2, S3, and S4), were identified and segmented by incorporating acoustical features of PCG signals with fiducials points of ECG. The waveforms of the heart sound segments were converted into the CABs of intensity, width, complexity [11], and strength [7]. The CABs within 10 s before and after the measurement timestamps for PAP and PCWP at baseline and after exercise were extracted and their mean values calculated for later analysis (Figure 1).

The three categories of CABs other than strength were geometrically derived as follows: Intensity, peak-to-peak amplitude on PCG inside the segments of heart sounds; Width, time duration on PCG inside the segments of heart sounds; and Complexity, the total areas of valleys created by the peaks in high-frequency components inside the segment. Strength is an exclusive category for abnormal heart sounds, S3 and S4. It is a probability score based on acoustic features reflecting the presence of an S3 and S4 for each recording. The parameter ranged from 0 to 10, and values above five indicated the existence of a clinically audible and persistent S3 or S4 in a 10-second recording. The scalograms in Figure 2 are visual descriptions of the change in S3 and of the corresponding S3 Strength in relation to hemodynamic stress caused by handgrip. S3 is usually subaudible (with a frequency below 20 Hz) [13] and visible right next to the S2 component in a scalogram. The S3 was observed intermittently at baseline (Figure 2a) and in every beat after exercise (Figure 2b). The salient and consistent presence of S3 leads to high S3 Strength. Although several studies evaluated the other CABs based on time intervals derived from fiducial points of ECG and PCG signal, e.g., electromechanical activation time and left ventricular systolic time [14,15], their associations with hemodynamic parameters have not been confirmed [16], and they were excluded from the analysis in this study.

### 2.3. Statistical Analysis

Numerical data are presented as means and standard deviation. Categorical data are indicated as numbers and percentages. The hemodynamic parameters (mean PAP and PCWP) and CABs (intensity, width, complexity, and strength) obtained from right heart catheterization were compared with time points using analysis of variance. The change values of each hemodynamic parameter and CAB were calculated, and their correlations were evaluated as absolute and change values from baseline to exercise using Pearson’s correlation coefficients and its 95% confidence interval.

Since this is an exploratory study, the sample size was not calculated for the purpose of evaluating the efficacy endpoint. The sample size was, therefore, determined to be 50 participants which is a feasible number to recruit at our hospital during a study period.

All patients for whom PAP values were obtained at baseline and after exercise were classified into increasing and decreasing response groups according to the change in mean PAP values. Fisher’s exact tests were used to compare PAP response in several subgroups of interest stratified by background factors at baseline, including median age, median body mass index, estimated glomerular filtration rate (60.0 mL/min/1.73 m^2^), median N-terminal pro-B-type natriuretic peptide, comorbidities, β-blocker use, heart failure status (left ventricular ejection fraction, E/e’, cardiac index, PCWP, and PAP), and pulmonary hypertension (PH; isolated post-capillary PH or combined post- and pre-capillary PH). All reported *p*-values were two-tailed, and *p* < 0.05 was considered significant. No adjustments for multiple comparisons were performed in this study. All statistical analyses were performed using R statistical software, version 4.0.2 (R Foundation for Statistical Computing, Vienna, Austria).

## 3. Results

### 3.1. Patient Demographics

PCG and ECG recordings could not be obtained for one of the 50 patients enrolled in the study; therefore, only 49 patients were included in the analysis dataset. Their baseline demographic and clinical characteristics are summarized in Table 1. At baseline, cardiac index and PVR was 2.20 ± 0.48 mL/m^2^, 183.55 ± 113.45 dynes·s·cm^−5^, respectively. The distribution of CABs and mean PAP, PCWP by the phases of right heart catheterization are described in Table 2. Mean PAP significantly increased, while mean PCWP did not. For the CABs, only S4 Intensity at the time of PAP measurement showed a significant difference.

### 3.2. Correlation between Hemodynamic Parameters and CABs

The correlation between absolute values of the hemodynamic parameters (mean PAP and PCWP) and CABs were summarized according to the phases of right heart catheterization, i.e., baseline and exercise, and are shown in Table 3 and Appendix A. Several CABs correlated significantly with mean PAP, among which S2 Width (*r* = 0.354; *p* = 0.014 and *r* = 0.363; *p* = 0.010) and S3 Strength (*r* = 0.375; *p* = 0.009 and *r* = 0.386; *p* = 0.007) were consistent throughout the phases. Unlike S3 Strength, S3 Intensity at baseline, but not after exercise, was correlated significantly with mean PAP (Appendix A). The S2 Width and S3 Strength also showed significant correlations with mean PCWP at baseline and after exercise (*r* = 0.308; *p* = 0.031 and *r* = 0.336; *p* = 0.019 for S2 Width, and *r* = 0.404; *p* = 0.004 and *r* = 0.353; *p* = 0.014 for S3 Strength).

The changes in CABs and hemodynamic parameters induced by handgrip exercise were also assessed using the correlation between the differences in the values (Table 4 and Appendix A). No combination indicated a significant association among the CABs correlated by absolute values in all phases of right heart catheterization.

### 3.3. Differences in Background Factors between Exercise-Induced Increases and Decreases in CABs

During exercise, mean PAP values increased (from baseline values) in all cases. The response of CABs to this change, however, was divided into two patterns: increasing and decreasing. Figure 3 shows the scatter plots of mean PAP and S3 Strength for both the increasing and decreasing patterns. In the figure, the baseline and exercise data points of the same case are connected by straight lines.

To compare patient background, S3 Strength at the time of mean PAP measurement was stratified into increasing and decreasing groups (Table 5). Among those subgroups, the frequency of cardiac index less than 2.2 mL/m^2^ was significantly higher in the decreasing group. The comparison of the background parameters between the increasing and decreasing groups in other CABs are summarized in Appendix A. The S3 Strength response to exercise-induced PCWP change differed significantly according to the presence or absence of left ventricular diastolic dysfunction (Appendix A). Regarding the diabetes mellitus comorbidity, S3 Strength showed a significant difference at the time of measurement of mean PCWP (Appendix A), whereas S2 Width at the time of measurement of mean PCWP did not show any significant background difference (Appendix A).

## 4. Discussion

This study examined the correlations between the absolute value of some CABs and hemodynamic parameters. Overall, S2 Width and S3 Strength showed significant correlations with PAP and PCWP, and these correlations were consistent at baseline and after exercise. S2 Complexity is reportedly significantly correlated with PAP and PVR in patients with PH [11,12], and those relationships were observed in the mean PAP after exercise in the present study within the HF population. S2 Width also correlated with mean PAP. S2 Width is the duration of S2 segmentation and generally becomes long when S2 is widely split. Some literature studies have reported S2 splitting to be an estimator of PAP; however, the results were inconsistent in terms of the relationship between S2 split width and PAP [17]. In this study, the positive correlation observed indicated that longer S2 Width can detect an elevation of PAP. S3 Strength is reportedly associated with mean PAP [12] and elevated PCWP [8] at rest. This is because increased hemodynamic load makes the third heart sound more salient, and that change is quantified by an increase in S3 Strength. The results of the present study are novel in the sense that those relationships were shown to be valid even under exercise load.

The correlations of CABs with hemodynamic parameters were thought to be enhanced when tested with changes in the parameters as reported in a comparison study with echocardiography during hospitalization [18]. However, the present result showed no significant correlation between the changes in CABs and the hemodynamic parameters as opposed to the result of absolute values. The reason for this difference may be inconsistencies in the responses of individual CABs to changes in hemodynamic conditions caused by the handgrip exercise. Since higher S3 Strength is reportedly associated with elevated filling pressure [8,9,10], it was anticipated that S3 Strength will increase with an increase in PAP. However, in the present study, S3 Strength decreased in response to increased mean PAP in 20 cases (42.5% of successful conversion). 

When comparing background factors with increasing and decreasing changes in S3 Strength at PAP measurement points, the decreasing group presented lower cardiac output, i.e., cardiac index less than 2.2 mL/m^2^, compared to the increasing group. Therefore, it was inferred that S3 Strength may respond to an elevation in PAP differently according to the HF condition and that cardiac output is possibly an influencing factor. Patients who had comorbid diabetes mellitus and possible diastolic dysfunction (E/e’ ≥ 14) also showed a significant difference in S3 Strength at PCWP measurement. Since there was no significant change in mean PCWP from baseline to exercise as a whole group, it was difficult to determine whether the changes in S3 Strength were directly caused by exercise load. In addition, the response direction of S3 Strength to an increase in PAP did not differ according to left ventricular diastolic function and actual PAP status. These results may imply that changes in S3 Strength are sensitively affected by low cardiac output but can be applied to evaluate diastolic dysfunction regardless of HF progression.

Early risk prediction of worsening HF can allow for appropriate interventions and prevent the occurrence of adverse events such as hospitalization [19,20]. Remote patient monitoring is a mainstream approach to predicting worsening HF outside hospitals and clinics [4,21,22] and continuous monitoring of PAP using implanted devices is reportedly effective in reducing the rehospitalization rate in patients with HF [3]. Noninvasive parameters reflecting hemodynamic status can be the key to making remote patient monitoring universally accessible to all patients with HF [23]. The correlation of S2 Width and S3 Strength with the invasive hemodynamic parameters in absolute values indicates that those CABs may help detect hemodynamic deterioration before the onset of HF symptoms. In contrast, the directions of changes in CABs in response to an elevation of PAP may differ based on the background hemodynamic status. This behavior implies that threshold-based risk prediction is ineffective without selecting target patients based on their baseline information.

This study has several limitations. First, the definition of HF in this study was based on the clinical manifestations only, and no cutoff values based on BNP or echocardiographic findings were specified. Thus, this study included a broad range of eligible patients with HF, possibly resulting in a heterogeneous study cohort. In particular, the study included patients with different HF phenotypes, some with reduced ejection fraction and some with mid-range/preserved ejection fraction, and from various clinical care settings, including acute and chronic phases of HF. However, the dataset was not large enough to allow a sub-analysis of the responsiveness of the parameters for each phenotype separately. Hence, further study is also required to assess the association between CABs and PAP in specific patient populations with different HF phenotypes. Second, this was a single-center study with a relatively small sample size. The background comparison of the increasing and decreasing groups in CABs might have been more accurate with a larger sample size. We need to accumulate cases to clarify background factors that affect the behavior of CABs for future remote monitoring. Third, PCWP did not change significantly at baseline and exercise, and this might have made the correlation of the parameter changes unclear. This could have been because the workload applied by handgrip was too weak or that by the upper body was too intensive to evoke a change in volume overload and thus PCWP. Lastly, the findings of this study are not applicable to patients with HF who have valve diseases (who were excluded from this study) such as severe aortic stenosis and degenerative mitral regurgitation. Heart murmurs, especially diastolic murmurs, may cause difficulty in the acoustical analysis of heart sounds and calculation of CABs.

## 5. Conclusions

CABs related to S2 and S3 showed significant correlations with absolute PAP values both at baseline and after exercise in patients with HF, but no significant correlations between their changes from baseline to post-exercise were observed in this study population. Further research is therefore needed to assess whether CABs can sensitively reflect changes in PAP according to HF status and underlying phenotypes.

## Figures and Tables

**Figure 1 jcm-11-06373-f001:**
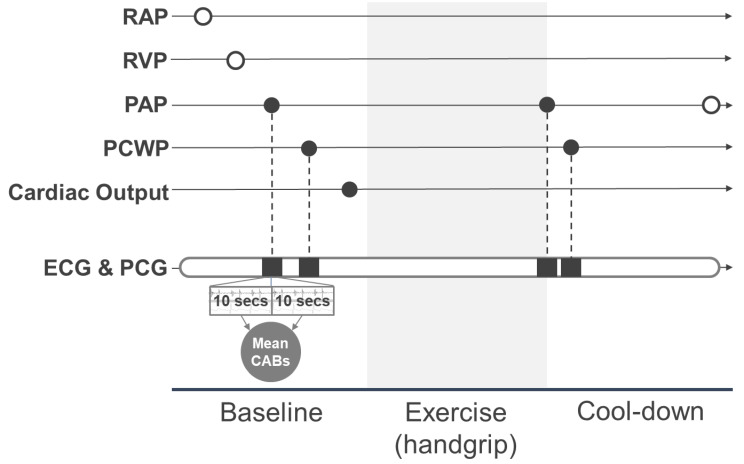
Measurement workflow for simultaneous recording of right heart catheterization and CABs. Filled circles represent the hemodynamic parameters used in the analysis. White circles represent the hemodynamic parameters not used in the analysis. RAP, right atrial pressure; RVP, right ventricular pressure; PAP, pulmonary artery pressure; PCWP, pulmonary capillary wedge pressure; ECG, electrocardiogram; PCG, phonocardiogram; CABs, cardiac acoustic biomarkers.

**Figure 2 jcm-11-06373-f002:**
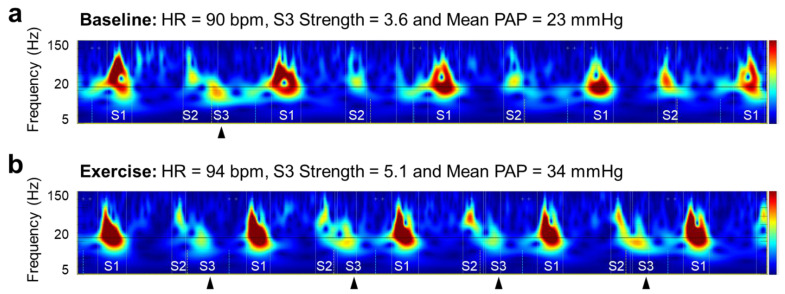
Visual description of change in S3 and corresponding S3 Strength in relation to hemodynamic alternation from baseline (**a**) to exercise (**b**). HR heart rate, PAP pulmonary artery pressure, S1 first heart sound, S2 second heart sound, S3 third heart sound.

**Figure 3 jcm-11-06373-f003:**
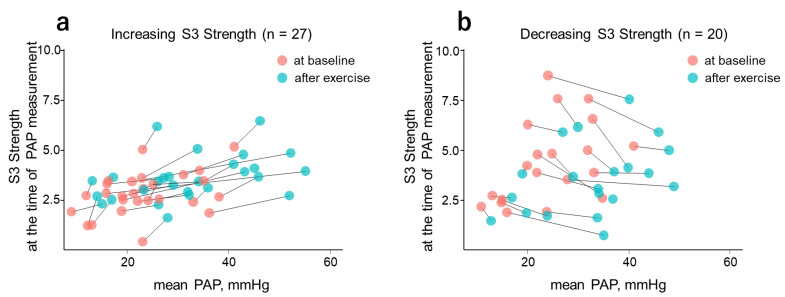
Scatter plot of mean PAP vs. S3 Strength at the time of PAP measurement. With the exercise-induced increase in mean PAP, S3 Strength showed increasing (**a**) and decreasing (**b**) tendencies. Light pink and blue circles represent measurement data at baseline and after exercise, respectively. S3, third heart sound; PAP, pulmonary artery pressure.

**Table 1 jcm-11-06373-t001:** Baseline clinical characteristics.

Variable	Overall (*n* = 49)
Age, year, mean ± SD	68.1 ± 11.2
Male, number (%)	37 (75.5)
BMI, kg/m^2^, mean ± SD	23.8 ± 4.3
eGFR, mL/min/1.73 m^2^, mean ± SD	52.6 ± 20.1
NT-proBNP, median [inter quartile range]	1353.0 [429.1, 2044.0]
LVEF, %, mean ± SD	36.3 ± 16.5
<40%, number (%)	32 (65.3)
40–49%, number (%)	7 (14.3)
≥50%, number (%)	10 (20.4)
E/e’, mean ± SD	13.8 ± 4.8
NYHA Functional classification, number (%)	
1	4 (8.2)
2	10 (20.4)
3	31 (63.3)
4	4 (8.2)
Etiology, number (%)	
Dilated cardiomyopathy	8 (16.3)
Ischemia	7 (14.3)
Arrhythmia	6 (12.2)
Valvular	5 (10.2)
Hypertensive	2 (4.1)
Amyloidosis	2 (4.1)
Sarcoidosis	1 (2.0)
Others/unknown	18 (36.7)
Implantable device, number (%)	5 (10.2)
Comorbidity, number (%)	
Coronary artery disease	5 (10.2)
Atrial fibrillation	20 (38.8)
Stroke	4 (8.2)
Diabetes mellitus	17 (34.7)
Hypertension	28 (57.1)
Medication, number (%)	
RAS inhibitor	41 (83.7)
Diuretic	39 (79.6)
β-blocker	34 (69.4)
Calcium channel blocker	8 (16.3)

Abbreviations: BMI, body mass index; eGFR, estimated glomerular filtration rate; NT-proBNP, N-terminal pro-B-type natriuretic peptide; LVEF, left ventricular ejection fraction; NYHA, New York Heart Association; RAS, renin-angiotensin system; SD, standard deviation.

**Table 2 jcm-11-06373-t002:** Distribution of hemodynamic parameters by right heart catheterization and of CABs by phases.

	At Baseline	After Exercise	*p* Value
Right Heart Catheterization			
Mean PAP, mmHg	23.52 ± 8.41	32.45 ± 11.28	<0.001
Mean PCWP, mmHg	15.39 ± 8.02	16.96 ± 7.96	0.335
CABs at PAP measurement			
HR, bpm	72.92 ± 13.05	75.93 ± 11.47	0.469
S1 Intensity, mV	7.68 ± 4.71	8.02 ± 4.13	0.704
S1 Width, ms	175.72 ± 33.92	168.36 ± 32.71	0.500
S1 Complexity	2.92 ± 0.94	2.98 ± 0.71	0.511
S2 Intensity, mV	6.17 ± 3.21	6.49 ± 2.97	0.829
S2 Width, ms	114.95 ± 21.14	111.77 ± 18.94	0.656
S2 Complexity	1.93 ± 1.04	2.05 ± 1.13	0.430
S3 Strength	3.51 ± 1.72	3.60 ± 1.42	0.849
S3 Intensity, mV	1.30 ± 0.46	1.39 ± 0.39	0.464
S4 Strength	3.93 ± 1.61	4.67 ± 1.93	0.221
S4 Intensity, mV	1.09 ± 0.46	1.54 ± 0.92	0.044
S2/S1 Intensity	0.93 ± 0.53	0.93 ± 0.49	0.916
S2/S1 Complexity	0.69 ± 0.36	0.73 ± 0.45	0.408
CABs at PCWP measurement			
HR, bpm	71.91 ± 11.44	74.06 ± 10.96	0.347
S1 Intensity, mV	7.56 ± 4.16	8.63 ± 4.87	0.248
S1 Width, ms	175.89 ± 33.33	171.94 ± 31.17	0.548
S1 Complexity	3.03 ± 0.91	2.75 ± 0.93	0.143
S2 Intensity, mV	6.25 ± 3.26	7.10 ± 4.10	0.262
S2 Width, ms	116.21 ± 23.74	110.85 ± 18.71	0.221
S2 Complexity	2.07 ± 1.22	1.66 ± 1.01	0.080
S3 Strength	3.75 ± 1.72	3.63 ± 1.58	0.720
S3 Intensity, mV	1.28 ± 0.39	1.82 ± 2.65	0.169
S4 Strength	4.20 ± 1.80	4.13 ± 1.98	0.892
S4 Intensity, mV	1.21 ± 0.82	1.42 ± 0.73	0.287
S2/S1 Intensity	0.96 ± 0.57	0.96 ± 0.56	0.959
S2/S1 Complexity	0.81 ± 0.84	0.62 ± 0.34	0.159

Abbreviations: CABs, cardiac acoustic biomarkers; PAP, pulmonary arterial pressure; PCWP, pulmonary capillary wedge pressure; HR, heart rate; S1, first heart sound; S2, second heart sound; S3, third heart sound; S4, fourth heart sound.

**Table 3 jcm-11-06373-t003:** Correlation between absolute values of PAP and CABs.

	Mean PAP			
	At Baseline		After Exercise	
	*n*	*r* (95% Confidence Interval)	*n*	*r* (95% Confidence Interval)
HR, bpm	48	0.212 (−0.077, 0.468)	49	0.159 (−0.127, 0.422)
S1 Intensity, mV	48	−0.153 (−0.419, 0.137)	49	−0.176 (−0.435, 0.111)
S1 Width, ms	48	−0.27 (−0.514, 0.016)	49	−0.189 (−0.446, 0.098)
S1 Complexity	48	−0.207 (−0.464, 0.082)	49	−0.251 (−0.497, 0.032)
S2 Intensity, mV	48	0.247 (−0.040, 0.496)	49	0.185 (−0.102, 0.443)
S2 Width, ms	48	0.354 (0.078, 0.580) *	49	0.363 (0.091, 0.584) *
S2 Complexity	48	0.249 (−0.038, 0.498)	49	0.312 (0.034, 0.545) *
S3 Strength	48	0.375 (0.102, 0.596) *	48	0.386 (0.114, 0.604) *
S3 Intensity, mV	47	0.335 (0.053, 0.568) *	46	0.270 (−0.022, 0.520)
S4 Strength	34	−0.075 (−0.403, 0.270)	29	−0.166 (−0.502, 0.214)
S4 Intensity, mV	34	0.109 (−0.238, 0.431)	27	−0.079 (−0.446, 0.31)
S2/S1 Intensity	48	0.296 (0.013, 0.535) *	49	0.259 (−0.024, 0.504)
S2/S1 Complexity	48	0.267 (−0.019, 0.512)	49	0.360 (0.087, 0.582) *

*: *p* < 0.05, *r*: correlation coefficient. Abbreviations: CABs, cardiac acoustic biomarkers; PAP, pulmonary artery pressure; HR, heart rate; S1, first heart sound; S2, second heart sound; S3, third heart sound; S4, fourth heart sound.

**Table 4 jcm-11-06373-t004:** Correlation between changes from baseline to post-exercise in PAP and CABs.

	Mean PAP	
	*n*	*r* (95% Confidence Interval)
HR, bpm	48	0.205 (−0.084, 0.462)
S1 Intensity, mV	48	0.009 (−0.276, 0.293)
S1 Width, ms	48	0.029 (−0.258, 0.310)
S1 Complexity	48	0.051 (−0.237, 0.330)
S2 Intensity, mV	48	−0.027 (−0.309, 0.259)
S2 Width, ms	48	−0.058 (−0.337, 0.230)
S2 Complexity	48	0.173 (−0.117, 0.436)
S3 Strength	47	0.089 (−0.204, 0.367)
S3 Intensity, mV	45	−0.005 (−0.299, 0.289)
S4 Strength	28	−0.057 (−0.421, 0.323)
S4 Intensity, mV	27	−0.033 (−0.408, 0.351)
S2/S1 Intensity	48	−0.011 (−0.294, 0.274)
S2/S1 Complexity	48	0.145 (−0.146, 0.412)

*r*: correlation coefficient. Abbreviations: CABs, cardiac acoustic biomarkers; PAP, pulmonary artery pressure; HR, heart rate; S1, first heart sound; S2, second heart sound; S3, third heart sound; S4, fourth heart sound.

**Table 5 jcm-11-06373-t005:** Comparison of background characteristics with S3 Strength response to exercise-induced PAP change.

			S3 Strength Response		
		Overall	Decreasing (*n* = 20)	Increasing (*n* = 27)	*p* Value
Age, number (%)					
(median)	<69 years	24 (51.1)	12 (60.0)	12 (44.4)	0.380
	≥69 years	23 (48.9)	8 (40.0)	15 (55.6)	
BMI, number (%)					
(median)	<24.0 kg/m^2^	24 (51.1)	12 (60.0)	12 (44.4)	0.380
	≥24.0 kg/m^2^	23 (48.9)	8 (40.0)	15 (55.6)	
eGFR, number (%)					
	<60.0 mL/min/1.73 m^2^	32 (68.1)	12 (60.0)	20 (74.1)	0.355
	≥60.0 mL/min/1.73 m^2^	15 (31.9)	8 (40.0)	7 (25.9)	
NT-proBNP, number (%)					
(median)	<1353 pg/mL	15 (51.7)	6 (50.0)	9 (52.9)	1.000
	≥1353 pg/mL	14 (48.3)	6 (50.0)	8 (47.1)	
Atrial fibrillation, number (%)					
	Yes	18 (38.3)	5 (25.0)	13 (48.1)	0.137
	No	29 (61.7)	15 (75.0)	14 (51.9)	
Diabetes mellitus, number (%)					
	Yes	16 (34.0)	6 (30.0)	10 (37.0)	0.758
	No	31 (66.0)	14 (70.0)	17 (63.0)	
Hypertension, number (%)					
	Yes	26 (55.3)	9 (45.0)	17 (63.0)	0.250
	No	21 (44.7)	11 (55.0)	10 (37.0)	
β-blocker use, number (%)					
	Yes	34 (72.3)	6 (30.0)	7 (25.9)	1.000
	No	13 (27.7)	14 (70.0)	20 (74.1)	
LVEF, number (%)					
	<40%	30 (63.8)	13 (65.0)	17 (63.0)	0.573
	40–49%	7 (14.9)	4 (20.0)	3 (11.1)	
	≥50%	10 (21.3)	3 (15.0)	7 (25.9)	
E/e’, number (%)					
	<14	29 (61.7)	11 (55)	18 (66.7)	0.546
	≥14	18 (38.3)	9 (45)	9 (33.3)	
Cardiac index, number (%)					
	<2.2 mL/m^2^	24 (51.1)	14 (70.0)	10 (37.0)	0.039
	≥2.2 mL/m^2^	23 (48.9)	6 (30.0)	17 (63.0)	
Mean PCWP, number (%)					
	<15 mmHg	23 (48.9)	8 (40.0)	15 (55.6)	0.380
	≥15 mmHg	24 (51.1)	12 (60.0)	12 (44.4)	
Mean PAP, number (%)					
	≤20 mmHg	17 (36.2)	7 (35)	10 (37)	1.000
	>20 mmHg	30 (63.8)	13 (65)	17 (63)	
PH, number (%)					
	Yes (Ipc-PH or Cpc-PH)	23 (51.1)	11 (57.9)	12 (46.2)	0.550
	No	22 (48.9)	8 (42.1)	14 (53.8)	

Abbreviations: PAP, pulmonary artery pressure; S3, third heart sound; BMI, body mass index; eGFR, estimated glomerular filtration rate; NT-proBNP, N-terminal pro-B-type natriuretic peptide; LVEF, left ventricular ejection fraction; PCWP, pulmonary capillary wedge pressure; Ipc, isolated post-capillary; Cpc, combined post- and pre-capillary; PH, pulmonary hypertension.

## Data Availability

The data presented in this study are available from the corresponding author upon request. The data are not publicly available due to confidentiality agreements with the research collaborators.

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
