# Peer review of "Relationship between Cardiac Acoustic Biomarkers and Pulmonary Artery Pressure in Patients with Heart Failure"

_jcm, 2022, doi:10.3390/jcm11216373_

Round 1

Reviewer 1 Report (Previous Reviewer 3)

All comments have been adequately addressed. I have no further considerations to make.

Author Response

Your comment: All comments have been adequately addressed. I have no further considerations to make.

Our response: Thank you so much for your kind review.

Reviewer 2 Report (New Reviewer)

As submitted, this is a single-center prospective proof-of-concept study that compares noninvasive measurement of acoustic biomarkers with the invasive acquisition of right sided cardiac pressures.  The authors identified 49 patients with heart failure and compared data generated from right heart catheterization with cardiac acoustic biomarkers obtained from phonocardiogram.  The authors conclude that certain CABs may correlate to PAP in heart failure patients.

Major reviewer comments:

1)  The heterogeneous and poorly described patient population that is presented in this study is problematic.  In 2022, it is most accurate to be able to fully flush out the differences between HFrEF and HFpEF.  Especially in this type of paper, one would even try and go so far as to also look at the HFmrPF subgroup.  This paper would greatly benefit from considering this added depth to their investigations.

-Though the authors do list the lack of subgroup analysis as a limitation of the paper, it is fair to put forth that this is a significant limitation that should not be buried in last few paragraphs of the discussion section.  

-Conversation about heart failure phenotypes are superficial in the background/introduction.  Accordingly, laying this limitation out for the reader at the beginning of the manuscript sets the tone for the data that is hoped to be conveyed.  

2)  Line 267 states that “Patients who had comorbid diabetes mellitus and diastolic dysfunction (E/e’ > 14) also showed a significant difference in S3 Strength at PCWP measurement.”  This line is confusing and misleading.  Echocardiographic criteria for the diagnosis of HfpEF (Pieske et al. Eur Heart J. 2019) extend far beyond “E/e’.”  Moreover, with RHC data available for all study subjects, one would believe that use of an echocardiographic surrogate marker for evaluating HFpEF is not warranted.   

3)  In discussion of the manuscript’s findings, it is mentioned that this paper’s conclusions stand in apposition to prior studies.  I wonder:  can the author’s provide a possible explanation for these discordant results?  Are there perhaps deeper physiological explanations for the results of this paper that could be shared with the reader?

Minor reviewer comments:

  1. Would include patients that are on a SGLT-inhibitor in your subgroup analysis in Tables 1 and 5 (and Tables S3-5 as well).
  2. The methods section needs a statement of how and where the wedge pressure was measured.
  3.  

Author Response

Thank you for your critical review.

Please see attached the response letter file.

Round 2

Reviewer 2 Report (New Reviewer)

Thank you for addressing the suggestions put forth

This manuscript is a resubmission of an earlier submission. The following is a list of the peer review reports and author responses from that submission.

Round 1

Reviewer 1 Report

The purpose of the manuscript is to use CABs from heart sound to quantify exercise-induced elevation of PAP in patients with HF. Authors have used data from 49 patients and used a correlation study to conclude that mean PAP significantly increased after exercise compared to baseline, several CABs correlated significantly with mean PAP such as S2 width, s3 strength, etc. Authors need to justify choosing heart sound in their experiment, the use of ECG and its effect in their experiment, rationale behind choosing CABs. 

Major points:

1.      Authors have recorded ECG for 49 patients, what is specifically the use of those 49 ECG recordings in analysis or their contribution to result/discussion?

2.      There was no significant change in mean PCWP from baseline to exercise, also authors have pointed out these as limitations that due to workload applied by handgrip was too weak. Also, the P value for mean PCWP was .335. – All these points toward the fact that PCWP should not be used as a CAB here for analysis or result or in any table to compare. Considering all these issues, It remains questionable why the authors kept/used PCWP in table 3, table 4? The authors need to clarify.

Minor points:

1.      In section 1, the authors have introduced their objective -  CABs during the elevation of PAP for patients with HF. Authors are suggested to spend a few sentences about the effectiveness of heart sound in the cause ( I saw reference 7 but it's not sufficient).

2.      “The waveforms of the heart sound segments were converted into the 117 CABs of intensity, width, complexity [8], and strength [7].” – Authors need to provide a rationale for choosing these four CABs.

3.      Based on the distribution of baseline clinical characteristics among patients, it's not an ideal scenario in terms of the dataset. Although authors have put that as a limitation but is quite possible that with a better/larger dataset, some or many of the authors concluded metrics will vary. Authors are suggested to consider the effect of the quality/size of the dataset ahead of preparing for any future experiment.

Author Response

Thank you so much for your review.

Reviewer 2 Report

The authors describe the correlation of heart sounds with hemodynamics obtained by phonocardiography.  The novel aspect of their study is the use of handgrip exercise to evaluate the correlation of changes in hemodynamics with changes in heart sounds.  Heart sounds were obtained by phonocardiography (AUDICOR AM-RT, Inovise Medical) and were recorded continuously during the study along with ECG.  Hemodynamics were made via right heart catheterization using standard fluid filled catheters and pressure transducers.  50 subjects were enrolled and acceptable recordings were obtained from 49 subjects.  This seems to be a reasonably sized study for the correlation of continuous variables, however, there is no mention of how sample size was determined.  I would presume this is a convenience sample size. 

The manuscript is well written and constructed.  The language used is clear. 

The description of how pressures were measured is not presented in detail, but given the broad use of these measurements in cardiovascular disease, this is reasonable.  Cardiac output was measured by the Fick principal using an assumed oxygen consumption.  Assumed oxygen consumptions can introduce substantial error into the measurement of cardiac output.  Cardiac output was only measured once during the study, at baseline.  PA pressures and PCW were measured at baseline and after handgrip exercise to exhaustion.  A final PA pressure is measured in recovery.  RA and RV pressures are measured at baseline only. 

The exercise intervention resulted in an increase in PA pressure (mean pressure at baseline 23.5 mmHg and after exercise 32.5 mmHg) and no change in other hemodynamic parameters.  There is a report of PVR before and after exercise, but one of two problems would occur with this measurement.  Either the PVR after exercise was calculated with the baseline cardiac output, which could be greatly changed from baseline following exercise, or the cardiac output following exercise was estimated with an assumed oxygen consumption, which also can be greatly in error.  The post-exercise PVR should be removed or somehow justified and the methods of calculating it explained. 

The correlations with heart sounds demonstrating a correlation of the S2 width with both mean PA pressure and mean PCW pressure.  There was also a correlation of S3 with both mean PA pressure and mean PCW pressure.  However, none of the heart sounds changed significantly with exercise.  Despite the negative results, the investigator chose to focus on their heart sound with the strongest correlation with PCW and PA pressures at rest, S3 magnitude.  They chose to divide the population into two groups, those in whom S3 increased in magnitude following exercise and those in whom S3 decreased in magnitude following exercise.  They do not provide a physiologic justification for dividing the population by this way, and the up and down trends in magnitude may be simply a matter of chance.  They go further and compare hemodynamics in the two groups, finding a narrowly statistically significant difference in cardiac index between these two groups.  There is no correction for multiple measurements, and the concern about the accuracy of the cardiac index measurement is already noted in this review. 

The conclusion of their study runs counter to their data.  Their statement, “CABs (cardiac acoustic biomarkers) related to S2 and S3 can quantify the acoustical characteristics of heart sounds which reflect an exercise-induced elevation in PAP in patients with HF,” is not accurate.  They do fairly present the lack of change in CABs with the exercise form they chose to study.  Unfortunately, that’s the final answer. 

Author Response

Thank you so much for your review.

Reviewer 3 Report

The paper of Kaneko, Tanaka et al. describes a novel approach in estimating PAP from heart sound characteristics analysed by phonocardiography.

They found that some cardiac acoustic biomarkers correlated significantly with PAP values.

The study has an interesting approach.

I have some comments to make:

- Please provide the x-y correlation plots for S3 strength, intensity and mean PAP values (the parameters of table 3).

- I don't seem to understand Table 4: are the values given for mPAP values change during exercise or the values after that? In that case I would recommend computing correlations with the deltas (i.e., the change between baseline and after exercise).

- Table 5: it's Atrial fibrillation, please correct the typo.

- Where there any patients with hypertrophic cardiomyopathy?

- Where there differences in beta - blocker medications?

- where there differences in renal function? BNP?

- Please explain why sometimes there were only 47 patients of the 48 in the tables. Readout failures? Insufficient data? Noise?

Author Response

Thank you so much for your review.

Round 2

Reviewer 1 Report

The authors have answered all the queries satisfactorily.